# Rational-Based Discovery of Novel β-Carboline Derivatives as Potential Antimalarials: From In Silico Identification of Novel Targets to Inhibition of Experimental Cerebral Malaria

**DOI:** 10.3390/pathogens11121529

**Published:** 2022-12-13

**Authors:** Fernanda de Moura Alves, Jessica Correa Bezerra Bellei, Camila de Souza Barbosa, Caíque Lopes Duarte, Amanda Luisa da Fonseca, Ana Claudia de Souza Pinto, Felipe Oliveira Raimundo, Bárbara Albuquerque Carpinter, Ari Sérgio de Oliveira Lemos, Elaine Soares Coimbra, Alex Gutterres Taranto, Vinícius Novaes Rocha, Fernando de Pilla Varotti, Gustavo Henrique Ribeiro Viana, Kézia K. G. Scopel

**Affiliations:** 1Research Center on Biological Chemistry (NQBio), Federal University of São João Del Rei, Divinópolis 35501-296, Brazil; 2Research Center Parasitology, Departament of Parasitology, Microbiology and Immunology, Federal University of Juiz de Fora, Juiz de Fora 36036-900, Brazil; 3Research Center of Pathology and Veterinary Histology, Departament of Veterinary Medicine, Federal University of Juiz de Fora, Juiz de Fora 36036-900, Brazil

**Keywords:** malaria, antimalarial chemotherapy, cerebral malaria, β-carbolines alkaloids, nitric oxide, experimental cerebral malaria

## Abstract

Malaria is an infectious disease widespread in underdeveloped tropical regions. The most severe form of infection is caused by *Plasmodium falciparum*, which can lead to development of cerebral malaria (CM) and is responsible for deaths and significant neurocognitive sequelae throughout life. In this context and considering the emergence and spread of drug-resistant *P. falciparum* isolates, the search for new antimalarial candidates becomes urgent. β-carbolines alkaloids are good candidates since a wide range of biological activity for these compounds has been reported. Herein, we designed 20 chemical entities and performed an in silico virtual screening against a pool of *P. falciparum* molecular targets, the Brazilian Malaria Molecular Targets (BRAMMT). Seven structures showed potential to interact with PfFNR, PfPK7, PfGrx1, and PfATP6, being synthesized and evaluated for in vitro antiplasmodial activity. Among them, compounds **3**–**6** and **10** inhibited the growth of the W2 strain at µM concentrations, with low cytotoxicity against the human cell line. In silico physicochemical and pharmacokinetic properties were found to be favorable for oral administration. The compound **10** provided the best results against CM, with important values of parasite growth inhibition on the 5th day post-infection for both curative (67.9%) and suppressive (82%) assays. Furthermore, this compound was able to elongate mice survival and protect them against the development of the experimental model of CM (>65%). Compound **10** also induced reduction of the NO level, possibly by interaction with iNOS. Therefore, this alkaloid showed promising activity for the treatment of malaria and was able to prevent the development of experimental cerebral malaria (ECM), probably by reducing NO synthesis.

## 1. Introduction

Malaria still continues scourge many countries in the world [1], having a long range in tropical underdeveloped regions, including countries in Africa, Asia, and South America, and causing approximately 241 million cases, which resulted in almost 627 thousand deaths in 2020 [2].

Although individuals from all ages are at risk of getting sick and dying from malaria, children under five years old compose the group that more frequently develop cerebral malaria (CM), the most severe form of the infection caused especially by *P. falciparum* [3]. The mechanisms driving the pathogenesis of CM are not fully known, but the sequestration of infected red blood cells (iRBC) and leucocytes in the brain causing capillary obstruction, inflammation, hemorrhage, and cerebral edema, which occur due to the disruption of the blood−brain barrier, play a critical role in the development of encephalopathy [4,5,6]. This syndrome is, therefore, the result of several mechanical events and a systemic inflammatory response with secretion of pro-inflammatory cytokines, including TNFα, IFNγ, and lymphotoxin. The cytokines, most notably TNFα, generate a vicious circle of macrophage activation, increasing the cellular sequestration and regulated production of reactive oxygen species, including nitric oxide (NO), in the general circulation and in situ [3,7]. Although 85% of subjects developing CM are able to make full recovery following the adequate antimalarial treatment, it is worryingly that about 25% of them remain with significant neurocognitive sequelae throughout life [8,9,10].

The current therapeutic arsenal for malaria treatment includes a combination of antimalarial drugs with diverse modes of action. Compounds like chloroquine (CQ) inhibit the hemoglobin polymerization in sexual stages of the parasites, while compounds like artemisinin impair more than one target by the destruction of parasite proteins due to the production of carbon centered radicals, resulting from metabolization of artemisinin (ART) into dihydroartemisinin [11]. However, the emergence and spread of resistant isolates of *P. falciparum* to current regimens of treatments makes it urgent to search for new antimalarial candidates that can contribute to the disease treatment, avoiding a worse prognosis [12,13,14].

β-carbolines are natural alkaloids, containing an attractive scaffold, that are found in many commercialized drugs [15] and antimalarial drug candidates [16]. These compounds show great chemical diversity and can be found as monomers or dimers. The saturation of the *N*-containing six-membered ring is used to classify the tricyclic monomers as β-carbolines for unsaturated members, and dihydro-β-carbolines (DHβC) and tetrahydro-β-carbolines (THβC) for partially saturated members. Therefore, a wide range of biological activity for these compounds have been reported [15,17].

The success in drug discovery requires the identification of molecules possessing optimal pharmacokinetic and pharmacodynamic properties. Over the years, medicinal chemists developed tools [18,19,20,21] that allow the selection of molecules containing safety attributes and druglike properties at the design stage, in order to increase the survival of drug candidates [22].

Herein, we describe the in silico guided choice and optimization studies, the synthesis and biological assays of new β-carboline derivatives with antimalarial activity aiming at a potential scaffold for antimalarial development.

## 2. Experimental Design 

### 2.1. Virtual Screening

An *in-house* library of 20 virtual chemical entities (Appendix A), which were possible to be synthesized within the commercially available reagents, was built to identify chemical entities displaying promising antiplasmodial activities using a virtual screening approach.

Twenty virtual chemical entities were designed using MarvinSketch^®^ software (ChemAxon, Cambridge, MA, USA); MOPAC^®^ software [23] was used to refine all molecular structures through the PM7 semi-empirical method; a EigenFollowing (EF) routine for geometry optimization and the charge was adjusted according with the protonation state in pH 7.4 [24,25]. Following, all structures were submitted to the molecular docking calculations by the AutoDock Vina^®^ program [26], by using the OCTOPUS^®^ platform, and the configuration files through a re-docking step were determined [27]. Thus, the virtual screening of potential antimalarial candidates was performed by using the BraMMT data bank [28]. Binding energy values, Δ (binding energy of the crystallographic ligand—binding energy of the compound) values, were calculated [29], and these values were employed as a first step in the selection process applied to choose the most promising structures to be synthesized.

### 2.2. Chemistry

Reagents and solvents were purchased as reagent grade from Sigma-Aldrich and used without further purification. Nuclear Magnetic Resonance (NMR) spectra were recorded using a Bruker Avance DRX-200 or DRX-400 and a Bruker AC300. Chemical shifts are reported as δ (ppm) downfield from tetramethylsilane (TMS) and the J-values reported in Hz. High-resolution mass spectrometry (HRMS) was recorded using an ESI micrOTOF-QII Bruker mass spectrometer. Microwave-assisted synthesis was carried out in a CEM Discover single-mode microwave reactor. Column chromatography was performed with silica gel 60, 70–230 mesh (Merck, Darmstadt, Germany).

#### 2.2.1. Synthesis of L-Tryptophan Methyl Ester (**2**)

Yield 92%, yellow oil to a suspension of commercial L-tryptophan (**1**) (2.45 mmol) in methanol (50 mL) concentrated sulfuric acid (2.81 mmol) was added dropwise. The reaction mixture was left under magnetic stirring for 48 h under reflux. The solution was cooled, neutralized with a saturated solution of sodium bicarbonate, and extracted with methylene chloride. The combined organic phases were dried (anhydrous Na_2_SO_4_), filtered, and then evaporated under reduced pressure. The residue obtained was filtered with ethyl acetate (70 mL) to yield the pure compound **2**. FT-IR (KBr) v/cm-1 3444, 3360, 3294, 2949, 1726, 1566, 1400, 1226, 1116. ^1^H NMR (CDCl_3_, 400 MHz) *δ* ppm: 2.95 (dd, *J* = 7.2 Hz, *J* = 14.3 Hz, 1H); 3.19 (dd, *J* = 4.4 Hz, *J* = 14.3 Hz, 1H); 3.59 (s, 3H); 3.74 (dd, *J* = 4.8 Hz, *J* = 7.3 Hz, 1H); 6.78 (s, 1H); 7.00–7.20 (m, 3H); 7.52 (d, *J* = 7.5, 1H). ^13^C NMR (50 MHz, CDCl_3_) *δ* ppm: 30.62; 51.92; 54.81; 111.22; 119.28; 121.91; 123.05; 110.64; 127.32; 136.22; 175.60.

#### 2.2.2. General Procedure for the Synthesis of **3**, **4**, **5**, **6**, **8**, and **9**

To a mixture of L-tryptophan methyl ester or tryptamine (0.30 mmol), aromatic benzaldehyde derivatives (0.90 mmol) in methylene chloride (35 mL) trifluoroacetic acid (TFA) (0.30 mmol) was then added dropwise. The resulting mixture was placed in a CEM Discover microwave oven (open vessel mode). Microwave irradiation of 100 W was used and the temperature ramped from 25 °C to 90 °C. Once 90 °C was reached, after around 7 min, the reaction mixture was held at this temperature for 30 min under stirring. All temperatures were measured externally by an IR sensor. After cooling the mixture, an aqueous solution of a saturated solution of sodium bicarbonate was added dropwise until pH 5, followed by extraction with EtOAc. The combined organic phases were dried (anhydrous Na_2_SO_4_), filtered, and then evaporated under reduced pressure. The residue was purified by flash chromatography (5:1:5 dichloromethane/ethyl acetate/hexane) to give the pure compounds **3**, **4**, **5**, **6**, **8**, and **9**.

#### 2.2.3. Methyl (1S,3S)-1-(2-Methoxy)-2,3,4,9-tetrahydro-1H-pyrido[3,4-b]indole-3-carboxylate (**3**)

Yield 81%, yellow solid. ^1^H NMR (DMSO, 400 MHz) *δ* ppm: 1.24 (s, 1H); 3.01–3.13 (m, 1H); 3.40–3.45 (m, 1H); 3.92 (s, 1H); 6.12 (s, 1H); 6.77-7.07 ( m, 4H); 7.13 (t, *J* = 7.7 Hz, 1H); 7.18 (d, *J* = 8.0 Hz, 1H ); 7.45 (t, *J* = 6.9 Hz, 1H); 7.54 (d, *J*=8.0 Hz, 1H); 10.96 (s, 1H). ^13^C NMR (DMSO, 100 MHz) *δ* ppm: 22.84, 49.13, 51.93, 53.41, 56.26, 106.66, 111.74, 118.62, 119.63, 120.87, 122.67, 125.76, 128.01, 130.51, 136.94, 157.66, 169.99. HRMS (ESI) *m/z* 337.1545[M + H]^+^ (theoretical 337.1546).

#### 2.2.4. Methyl (1R,3S)-1-(2-Methoxy)-2,3,4,9-tetrahydro-1H-pyrido[3,4-b]indole-3-carboxylate (**4**)

Yield 11%, white solid. ^1^H NMR (DMSO, 400 MHz) *δ* ppm: 1.21 (s, 1H); 2.83 (t, *J* = 10.9 Hz, 1H); 3.00–3.04 (m, 1H); 3.66 (s, 3H); 3.85 (s, 3H); 3.90 ( s, 1H); 5.68 (s, 1H); 6.88-7.02 (m, 4H); 7.10 (t, *J* = 7.8 Hz, 1H); 7.22 (d, *J* = 7.8 Hz, 1H); 7.31(t, *J* = 7.0 Hz, 1H); 7.42(d, *J* = 7.5 Hz, 1H); 10.26 (s, 1H). ^13^C NMR (DMSO, 100 MHz) *δ* ppm: δ 25.42, 52.5, 55.81, 56.28, 107.17, 111.42, 117.69, 118.66, 120.70, 120.92, 126.74, 129.04, 129.16, 136.35, 157.37, 173.22. HRMS (ESI) *m/z* 359.1349[M + Na]^+^ (theoretical 359.1474).

#### 2.2.5. Methyl (1S,3S)-1-(2-Chloro)-2,3,4,9-tetrahydro-1H-pyrido[3,4-b]indole-3-carboxylate (**5**)

Yield 32%, yellow solid. ^1^H NMR (DMSO, 400 MHz) *δ* ppm: 1.99 (s, 1H); 2.89 (t, *J* = 11.2 Hz, 2H); 3.69 (s, 3H); 3.94 (d, *J* = 9.7 Hz, 1H); 5.68 (s, 1H); 6.97 (t, *J* = 7.0 Hz, 1H); 7.02 (t, *J* = 7.0 Hz, 1H); 7.22 (d, *J* =7.8 Hz, 2H); 7.29 (t, *J* = 7.3 Hz, 1H); 7.36 (t, *J* = 7.3 Hz, 1H); 7.45 (d, *J* = 7.5 Hz, 1H); 7.53 (d, *J* = 7.8 Hz, 1H); 10.44 (s, 1H). ^13^C NMR (DMSO, 100 MHz) *δ* ppm: 25.24, 51.79, 56.04, 107.61, 111.27, 117.61, 118.50, 120.89, 126.46, 127.41, 129.42, 133.26, 134.13, 136.35, 172.86. HRMS (ESI) *m/z* 341.1044 [M + H]^+^ (theoretical 341.1051).

#### 2.2.6. Methyl (1R,3S)-1-(2-Chloro)-2,3,4,9-tetrahydro-1H-pyrido[3,4-b]indole-3-carboxylate (**6**)

Yield 40%, white solid. ^1^H NMR (DMSO, 400 MHz) *δ* ppm: 1.99 (s, 1H); 3.09 (t, *J* = 10.5 Hz, 2H); 3.33 (s, 3H); 3.69-3.72 (m, 1H); 5.72 (s, 1H); 6.79 (t, *J* = 7.5 Hz, 1H); 6.99 (t, *J* = 7.0 Hz, 1H); 7.05 (t, *J* = 7.5 Hz, 1H); 7.23 (d, *J* = 7.5 Hz, 2H); 7.32 (t, *J* = 7.8Hz, 1H); 7.48 (d, *J* = 7.5 Hz, 1H); 7.53 (d, *J* = 7.8 Hz, 1H); 10.70 (s, 1H). ^13^C NMR (DMSO, 100 MHz) *δ* ppm: 24.86, 51.04, 51.42, 51.74, 107.83, 111.14, 117.72, 118.46, 121.06, 126.36, 126.87, 129.14, 129.63, 130.16, 132.85, 133.20, 136.17, 139.62, 173.52. HRMS (ESI) *m/z* 362.9270 [M + Na] ^+^ (theoretical 363.0979).

#### 2.2.7. 5-(Diethylamino)-2-(2,3,4,9-tetrahydro-1H-pyrido[3,4-b]indol-1-yl)phenol (**8**)

Yield 60%, yellow solid. ^1^H NMR (DMSO, 400 MHz) *δ* ppm: 1.08 (t, *J* = 6.8 Hz, 6H); 1.26 (s, 1H); 3.00 (t, *J* = 7.9 Hz, 2H); 3.33 (q, *J* = 7.9 Hz, 2H); 3.75 (t, *J* = 6.8 Hz, 4H); 5.89 (s, 1H); 6.13 (d, *J* = 8.6 Hz, 1H); 7.02 (m, 5H); 7.34 (d, *J* = 7,8 Hz, 1H); 7.58 (d, *J* = 7.8 Hz, 1H); 8.13 ( s, 1H); 10.83 (s, 1H). ^13^C NMR (DMSO, 100 MHz) *δ* ppm: 12.59, 26.87, 43.74, 56.66, 102.65, 107.80, 111.34, 111.62, 118.23, 118.32, 120.88, 122.92, 127.14, 133.04, 136.19, 151.1. HRMS (ESI) *m/z* 336.2069 [M + H]^+^ (theoretical 336.1998).

#### 2.2.8. 1-(3,4,5-Trimethoxyphenyl)-2,3,4,9-tetrahydro-1H-pyrido[3,4-b]indole (**9**)

Yield 84%, white solid. ^1^H NMR (DMSO, 400 MHz) *δ* ppm: 3.00 (s, 1H); 3.12 (t, *J* = 7.8 Hz, 2H); 3.58 (t, *J* = 12.4 Hz, 2H); 3.74 (s, 9H); 5.59 (s, 1H); 6.75 (s, 1H); 7.05 (t, *J* = 7.1 Hz, 1H); 7.13 (t, *J* = 7.6 Hz, 1H); 7.32 (d, *J* = 8,1 Hz, 1H); 7.54 (d, *J* = 7.8 Hz, 1H); 10.89 (s, 1H). ^13^C NMR (DMSO, 100 MHz) *δ* ppm: 18.20, 40.50, 56.04, 60.03, 107.15, 111.62, 118.28, 119.09, 122.04, 125.70, 128.41, 129.91, 136.59, 153.10. HRMS (ESI) *m/z* 339.1703 [M + H]^+^ (theoretical 339.1630).

#### 2.2.9. 1-(3,4,5-Trimethoxyphenyl)-9H-pyrido[3,4-b]indole (**10**)

Yield 21%, white solid. To a cooled solution of tetrahydro-β-carbolines **9** (0.50 mmol) in DMF (5 mL) at 0 °C, potassium permanganate (0.80 mmol) was added. The mixture was kept under magnetic stirring at room temperature for 3 h. After, the reaction mixture was filtered and diluted with ethyl acetate (25 mL). The organic phase was washed with brine (25 mL), dried (anhydrous Na_2_SO_4_), and then evaporated under reduced pressure. The residue was purified by flash chromatography (4:6 ethyl acetate/hexane) to give the pure compound **10**: ^1^H NMR (DMSO, 400 MHz) *δ* ppm: 3.76 (s, 3H); 3.92 (s, 6H); 7.26 (t, *J* = 7.3 Hz, 3H); 7.55 (t, *J* = 7.0 Hz, 1H); 7.65 (d, *J* = 8.3 Hz, 1H); 8.10 (d, *J* = 5.1 Hz, 1H); 8.27 (d, *J* = 7.8 Hz, 1H); 8.44 (d, *J* = 5.1 Hz, 1H); 11.52 (s, 1H). ^13^C NMR (DMSO, 100 MHz) *δ* ppm: 56.07, 60,29, 105.88, 112.64, 113.99, 119.71, 121.09, 121.83, 128.33, 133.17, 134.09, 138.02, 138.38, 141.25, 153.27. HRMS (ESI) *m/z* 335.1394 [M + Na]^+^ (theoretical 335.1389).

### 2.3. Antimalarial Activity

#### 2.3.1. Antiplasmodial Activity

*Plasmodium falciparum* W2 strain (CQ-resistant) (MRA-157) was obtained from Malaria Research. Continuous culture was maintained as previously described [30,31]. In vitro antiplasmodial activity of the seven compounds against the *P. falciparum* culture was evaluated using the SYBR test as previously described [32]. Compounds were diluted to concentrations ranging from 10 to 0.0001 μM. Chloroquine (CQ) and artemisinin (ART) were used as reference antimalarial drugs. The plate was read in a fluorometer (Fluoroskan Ascent, Thermo Laboratories) with excitation at 485 nm and an emission of 535 mm. All experiments were performed in triplicate. The results were expressed as the mean of minimum lethal dose for 50% of the culture (IC_50_).

#### 2.3.2. In Vitro Cytotoxicity on Mammalian Cells

In vitro cytotoxicity of the compounds was assessed against human pulmonary fibroblast cells WI-26VA4 (ATCC CCL-95.1, USA), by 3-(4,5-dimethyl-2-thiazolyl)-2, 5-diphenyl-2H-tetrazolium bromide (MTT assay) [33]. Cells were cultured in a RPMI-1640 medium supplemented with 10% heat-inactivated fetal bovine serum in a 96-well plate. Compounds were diluted in different concentrations ranging from 100 to 1 μM and incubated with the cells for 48 h in a 5% CO_2_ atmosphere at 37 °C. The optical density was determined at 540 nm to measure the signal and background, respectively (Spectra Max340PC384, Molecular Devices, Sunnyvale, CA, USA). The minimum lethal dose for 50% of the cells (IC_50_) was determined as previously described [34].

#### 2.3.3. Selectivity Index (SI)

Selectivity index (SI) is the ratio between the cytotoxic and antiplasmodial activities of each compound tested. The values greater than 10 were considered indicative of lack of cytoxicity, whereas the substances with values below 10 were considered toxic [35].

### 2.4. Evaluation of Physicochemical and Pharmacokinetic Properties

Physicochemical properties of all chemical entities drawn were in silico analyzed according to Lipinski’s rule of 5 using the SwissADME website [36]. The properties of molecular mass (MW), topological polar surface area (TPSA), *consensus* partition coefficient (ClogP), number of hydrogen bond donors (HBD), and number of hydrogen bond acceptors (HBA) were predicted.

The absorption pharmacokinetics were estimated by the AdmetSar website [37] through the predictive data of absorption to the human intestinal epithelium (HIE), blood–brain barrier (BBB), and CACO-2 cell permeability [38,39].

### 2.5. Evaluation of In Vivo Antimalarial Activity

#### 2.5.1. Animals

C57BL/6 female mice, 6 to 8 weeks of age, from the Center of Reproduction Biology (CBR)-CIAEP 02.0048.2019-of the Federal University of Juiz de Fora—UFJF were used in the in vivo experiments, after approval by the ethics committee on animal research (UFJF, protocol #022/2020). The animals (n = 5/cage) were housed in ALESCO^®^ climate-controlled cabinets with controlled air flow, a temperature of 22 ± 2 °C and light control, respecting the animals’ circadian rhythm (light/dark cycle of 12 h), with standard rodent food and water ad libitum.

All experiments were carried out seeking to minimize the suffering of the animals. The care of the laboratory animals and all experimental animal operations was performed according to the international standards recommended in the Manual on the care and use of laboratory animals [40] and the ethical precepts for animal experimentation defined by the National Council for the Control of Animal Experimentation [41].

#### 2.5.2. Mouse Model of ECM

To carry out the proposed experiments, the murine model of ECM was used to assess the antimalarial activity of the compounds. *Plasmodium berghei* ANKA was kindly provided by Claudio Romero Marinho (Federal University of São Paulo-Brazil), and was used to conduct the infections which were performed by intraperitoneal (i.p.) injection of 10^5^ parasitized red blood cells (pRBCs) obtained from infected donor mice, as previously described [42,43]. Reduced responsiveness to external stimuli, ataxia, paralysis, coma, and/or convulsions were considered clinical signals of ECM [44].

#### 2.5.3. Drug Administration

After infection, mice were randomized into seven groups: (1) PbA (Infected), (2) CQ treated (positive control), and five treated groups with 10 mg/kg of β-carboline compounds. In the curative test, the mice were treated by oral gavage starting from day 3 (when the first PbA asexual blood stages were observed in the blood film) until day 7 post infection (p.i.). In the suppressive modified test [45], mice were treated daily by oral gavage for five consecutive days, starting four hours after infection.

#### 2.5.4. Basic Indicator Evaluation

Parasitemia was monitored using giemsa-stained blood smears and the inhibition of parasite multiplication (IPM) was calculated by the formula: [(A–B)/A] × 100, where A = parasitemia of the PbA, and B = parasitemia of the group treated with the compounds. The clinical symptoms were evaluated using the rapid murine coma and behavior scale (RMCBS) from day 5 p.i. [44] being maintained until the end of experimentation. The RMCBS consists of 10 parameters (gait, balance, motor performance, body position, limb strength, touch escape, pinna reflex, toe pinch, aggression, and grooming). Each parameter is scored 0 to 2, with 0 being the lowest function and 2 being the highest. Animals with clinical score ≤ 5 were euthanized. The survival rate (SR) was calculated based on the number of alive animals after 14 days of clinical evaluation.

#### 2.5.5. Macroscopic and Histological Analysis of the Brain

Mice presenting clinical scores ranging from 0–5 were anesthetized with an association of xylazine 1% and ketamine 5%, i.p. Macroscopic brain damage, such as hemorrhages, was evaluated by complete removal of the skin on the upper portion of the animals’ heads. After photographic recording, the animal was euthanized following cardiac perfusion with PBS to remove non-adhered RBC. Animals that did not present clinical scores compatible with neurological damage were followed up until the 14th day of infection, when they were euthanized according to the experimental procedures described above.

For histological analysis, the brain was dissected and routinely processed for fixation in formaldehyde 10% for 24 h and paraffin embedding. Sections of 5 µm were stained by Hematoxylin-Eosin (HE). The histopathology analysis of the mice brains was performed under an optical microscope (AX10, Zeiss, Oberkochen, Germany) in 40× and 100× magnification lenses to observe, qualitatively, the occurrence of areas of hemorrhages and vascular obstruction [46].

### 2.6. Evaluation of the In Silico and In Vitro Inhibition Potential of Nitric Oxide Synthesis

#### 2.6.1. In Vitro Cytotoxicity against Mammalian Cells

Macrophages obtained from the peritoneal cavity of BALB/c mice previously inoculated with 3% thioglycollate medium for the evaluation of cell viability were used. The macrophages were seeded in culture bottles with a RPMI-1640 medium supplemented with 2 mM L-glutamine, 100 µg/mL antibiotics (streptomycin and penicillin), 5% fetal bovine serum (FBS) and kept in an incubator at 5% CO_2_ atmosphere at 37 °C until the day of the test. The protocol was approved by the Ethical Committee for Animal Research of Federal University of Juiz de Fora (n° 007/2018-CEUA).

Cytotoxicity was assessed by cell viability using the MTT method [47]. Briefly, the macrophages at 2 × 10^6^ cells/well were transferred to 96-well microplates and treated with compound **10** at concentrations of 18 to 150 μM. For the negative control, cells were treated only with 0.06% DMSO (vehicle). The cells were incubated for 48 h at 37 °C and 5% CO_2_ atmosphere. Absorbance was read at 570 nm and cell viability (%) of macrophages was obtained by the following formula: % viable cells = [(AUSample)/AU0.06% DMSO] × 100, where: AUSample: Absorbance was measured after treatment of cells with the sample; AU0.06% DMSO: Absorbance was measured after treatment of cells with 0.06% DMSO. The experiment was performed in triplicate.

#### 2.6.2. Evaluation of Anti-Inflammatory Activity of Compound **10** by NO Dosage

NO production was determined in the culture supernatants of macrophages after being incubated for 48 h in the presence of compound **10** (18 to 150 μM/mL) and stimulated with LPS at 1 μg/mL and IFN-γ at 1 ng/mL. Negative control was stimulated with LPS and IFN-γ and treated with 0.06% DMSO (vehicle). The cells were incubated for 48 h at 37 °C and 5% CO_2_ atmosphere. Subsequently, NO production was evaluated by the Griess method (an indirect NO measurement by nitrite dosing) in the recovered culture supernatant from the 48 h stimuli and treatment as described by [48]. The quantity of NO (μM) was determined by comparison with a standard sodium nitrite solution. The spectrophotometric measurement was performed at 540 nm and the experiment was performed in triplicate.

#### 2.6.3. Evaluation of the Interaction with iNOS by Molecular Docking Simulations

The 2D structure of compound **10** was drawn using Chem Sketch software^®^ and their structures optimized in the Avogadro software v1.1 using the MMFF94s force field and the steepest descent algorithm. In the second step, the ligands were prepared in AutoDock Tools (ADT) v1.5.6, where the partial Gasteiger charges were added, rotatable bonds and hydrogens were set, and pdbqt files were generated. The iNOS enzyme (3EAI.pdb file) was used as a receptor and the compound as flexible ligands [49]. The three-dimensional structure of iNOS was downloaded from the RCSB Protein Data Bank and had its structure prepared in AutoDock Tools, where the hydrogens and Gasteiger charges were added and the pdbqt file was generated. The grid box used was: center box with dimensions of 22 Å × 22 Å × 22 Å; coordinates of X = 126.798, Y = 115.912, and Z = 89.696; and spacing of 0.375 Å. Molecular docking was performed with AutoDock Vina [26]. The program gave numerous feasible docked models (9 poses) that were evaluated in Pymol 2.4.1^®^ software. The most plausible structure based on the energetic parameters and interactions formed between the iNOS and ligands was selected.

#### 2.6.4. Statistical Analysis

For the statistical analysis, the Shapiro-Wilk normality test (n < 30 samples) was performed, followed by an analysis of variance by OneWay ANOVA and Tukey post-test for multiple comparisons. Analyses were performed using the GraphPad Prism software version 5.0 for Windows (GraphPad Software). Statistical significance was defined at the 5% level (*p* < 0.05).

## 3. Results and Discussion

### 3.1. Virtual Screening Results

For identification of possible targets of 20 β-carbolines compounds (Figure 1 and Appendix A), in silico simulations using molecular docking against BRAMMT were performed. The targets that exhibited binding energy with compounds lower than the crystallographic ligands are presented in Table 1. 

The 2OK8 model is a *P. falciparum* ferredoxin-NADP+-reductase (PfFNR) located in the apicoplast. 2OK8 transfers a pair of electrons to the iron–sulfur protein-ferredoxin (Fd) [50]. The PfFNR/Fd pair participates in several biosynthetic pathways in the apicoplast, performing electron transfer from NADPH to proteins in fatty acid synthesis pathways and mevalonate-independent isoprenoid pathways [51,52].

Inhibition of PfFNR interrupts the activity of HMB-PP reductase (IspH), consequently reducing the synthesis of dimethylallyl pyrophosphate (DMAPP) and isopentenyl pyrophosphate (IPP) and their derivatives, leading to the death of the protozoan [51]. These results reinforce the hypothesis that β-carbolines would act by inhibiting the synthesis of isoprenoids in malaria parasites [16,53,54,55,56].

The results also indicate a possible inhibition of other targets of the parasite, suggesting that the compounds could be acting in the inhibition of more than one metabolic pathway. Among the other targets, proteins such as PfPK7 (2PML model) [57], an important kinase for development and proliferation of the parasite, PfGrx1 (4N0Z model) [58], which acts in the oxidative stress control pathways, and PfATP6 [59], responsible for intracellular calcium ion homeostasis, could be involved in the action mechanism of the β-carbolines.

To evaluate a possible profile of interaction, an analysis of the interaction profile of β-carbolines derivatives against 2OK8 was performed. Compound **3** (Figure 1 and Appendix A) exhibited two hydrogen bonds, one between the nitrogen of the pyridine ring with Asp301 residue, and another between the oxygen attached to the D-ring and a Lys307 residue. It also demonstrated a Pi-Anion bond between the D-ring and Asp301. Similarly, compound **4** (Figure 1 and Appendix A) presented a hydrogen bond and a Pi-Anion bond with Asp301 residue.

The complex formed between compound **8** and the 2OK8 model (Figure 1 and Appendix A) forms three Pi-Anion bonds and one Alkyl bond with a Lys278 residue, one Alkyl bond with Arg310, one carbon hydrogen bond and Pi-Anion bond with a Lys306 residue, and a Pi-Alkyl bond with a Lys307. Although, there was an unfavorable binding between the nitrogen of the pyridine ring and Lys306. The greater number of bonds presented by compound **8** was compensatory for the reduction of binding energy. Unlike the other compounds tested, the amine linked to the D-ring is a longer substituent, which promotes more non-polar bonds, but changes the ligand conformation to prevent interaction with Asp301, as observed in the other derivatives.

Compound **5** (Figure 1 and Appendix A) presented only two bonds with the Asp301 residue, a hydrogen and a Pi-Anion bond, maintaining the same pattern of the other tested derivatives. Compound **6** (Figure 1 and Appendix A) forms, in addition to the bonds with the Asp301 residue, a bond between the chlorine substituent of the D-ring and an Asn275 residue, in addition to a hydrogen carbon bond with this same residue.

Compound **9** (Figure 1 and Appendix A) exhibited bonds with residues Asp301, Gln302, Lys307, and Asn276, in addition to the Pi-Anion bond with Asp301, which was common to all derivatives. Bonds with the Asp301 residue were also maintained in compound **10** (Figure 1 and Appendix A), in addition to a hydrogen bond between the Asn272 residue and the oxygen attached to the D-ring.

The analysis of the interaction profiles of the tested compounds showed that in all cases, the interaction of the Asp301 residue was present and promoted the greatest reduction in the binding energy, suggesting that the inhibition of PfFNR would occur through the interaction between the β-carbolines moiety and that aspartate residue.

### 3.2. Synthesis

The synthetic route for the β-carboline derivatives **3**–**6** and **8**–**10** is outlined in Figure 1. In the first step, the commercially available L-tryptophan was converted to L-tryptophan methyl ester in an esterification reaction using sulfuric acid and methanol. The L-tryptophan methyl ester previously synthesized and the commercially available tryptamine were subjected to a Pictet–Spengler reaction to obtain tetrahydro-β-carboline derivatives **3**–**6**, **8** and **9** upon reaction with different aromatic benzaldehydes in the presence of trifluoroacetic acid and chloroform as a solvent under microwave irradiation. The oxidation of **9** with potassium permanganate and DMF as a solvent at room temperature afforded compound **10.**

### 3.3. Biological Evaluation

#### 3.3.1. Cytotoxicity on Mammalian Cells and Antiplasmodial Activity

In vitro cytotoxicity of the 07 compounds was evaluated on human cell fibroblast lineage (WI-26-VA4 # ATCC CCL-95.1). None of the compounds were cytotoxic up to the highest concentration used (100 μM) (Table 2). The antiplasmodial activity assessment, conducted against chloroquine-resistant *P. falciparum* (W2 strain), showed IC_50_ values ranging from 0.51 to 1.82 μM (Table 2). It is interesting to point out that the compound **10** showed the beter antiplasmodial effect (IC_50_ = 0.51 μM) than CQ (IC_50_ = 0.59 μM). Regarding the SI, all compounds showed high selectivity for the parasites (SI > 10) (values ranged from >55 to >196) (Table 2).

The antiplasmodial activity of compounds containing the β-carboline scaffold have been reported over the years [16,55,56,60,61]. The reported in vitro antiplasmodial activity of all compounds reaches a recently established set of criteria for antimalarial hits, which includes: knowledge of the structure-activity; an inhibitory concentration half-maximum response (IC_50_) < 1 μM; and an SI greater than 10-fold against a human cell line [62]. 

#### 3.3.2. Evaluation of In Silico Physicochemical and Pharmacokinetic Parameters

Due to the low IC_50_ values and high selectivity, we decided to evaluate the in silico prediction of physicochemical and pharmacokinetic parameters of absorption to assess the bioavailability of compounds **3**–**6** and **10** when administered orally. As oral is the main route of drug administration, it is imperative that a drug candidate presents good bioavailability. The compound needs to be well absorbed to become available in the bloodstream and be delivered to the site of action [20,63,64].

The absorption of a drug can be predicted through its physicochemical properties, and the Lipinski’s rule of 5 is an important filter in the evaluation of these properties that predict whether the compound will present good bioavailability when administered orally. According to this rule, the compound must have molecular weight ≤ 500 g/mol, octanol/water partition coefficient (Log *p*) ≤ 5, number of hydrogen donors (NH + OH) ≤ 5, and number of hydrogen acceptors (N + O) ≤ 10 [18].

The SwissADME online platform provides information about the physicochemical properties of a compound through its molecular structure or the molecule’s simplified molecular input line entry system (SMILES) notation. All five compounds selected in the study presented LogP < 5 and MW < 500, featuring a good hydrophilic-lipophilic balance and molecular weight within the ideal range to allow adequate distribution and absorption by the plasma membranes (Table 3). In addition, all compounds showed a number suitable from donors to acceptors of hydrogen bonds, according to Lipinski’s Rule of 5, which suggests good oral bioavailability.

When an orally administered drug dissolves in the gastrointestinal tract, it must be sufficiently permeable through the biological membranes present to enter the systemic circulation. Since all compounds were active in the in vitro chemotherapy experiments, we proceeded with their pharmacokinetic analysis using the AdmetSar 2.0 software [37]. As observed in the Table 3, all selected compounds presented TPSA < 140 Å^2^, which is predictive of a good absorption [38,39]. Corroborating this information, the in silico prediction of the compounds showed high permeability in Caco-2 cells, intestinal epithelium, and the blood–brain barrier, suggesting that the compounds have a high probability (>95%) of being absorbed by the human intestine, reaching the systemic circulation, and being able to cross the blood–brain barrier. Since the analysis of the ADME properties of β-carboline alkaloids suggests good oral bioavailability, as well as good absorption by biological membranes, presenting a pharmacokinetic profile like CQ, all compounds were selected for in vivo assays of antiplasmodial activity.

#### 3.3.3. In Vivo Antimalarial Activity

For the in vivo assays, the curative experiment was initially carried out, in which the compounds **3**, **4**, **5**, **6,** and **10** were tested orally (10 mg/kg) for five days, starting the treatment on the 3rd dpi, after the appearance of the first iRBC. The recommendation for in vivo efficacy is that a lead compound should achieve parasite clearance at a dose that eradicates 90% of the target pathogen when administered orally (<50 mg/kg) in the blood stages of infection (typically four doses in 4 days) in the severe malaria mouse model [62].

As shown in Table 4, the mean parasitemia of infected mice and treated with compounds **4**, **6**, and **10**, was statistically lower than the observed on the untreated group on the 5th dpi (Tukey test, *p* < 0.05). Likewise, no difference was observed in the proportion of circulating parasites in the treated groups with alkaloids **4**, **6**, and **10** related to the standard antimalarial at 5th dpi (Tukey Test, *p* > 0.05). However, on the 7th dpi, the mean parasitemia observed for all alkaloids tested was equivalent to that observed for the untreated group (Table 4), suggesting loss of efficacy.

C57BL/6 mice infected with PbA, without any treatment, develop neurological signs (reduced exploratory behavior, decreased reflex, self-preservation, coma, and epilepsy) resulting in death between days 6 and 11 p.i. [42,43,65], data consistent with results obtained in this study (Figure 1A–C). However, the animals treated with compound **4** were resistant to the development of CM, as 90% of them remained alive until the 14th dpi (Figure 1A–C). In contrast, only 30% of animals treated with its isomer survived until the 14th dpi without demonstrating any CM signal. Treatment with compound **10** was also able to prevent the development of CM in 70% (Figure 1A–C) of the animals. No protective effect against CM development was observed when the animals were treated with compounds **5** and **6**. In Figure 1D,E, it is possible to observe extensive areas of cranial hemorrhage in the animals that were not protected from CM, following or not the treatment with compounds. Consistently with clinical assessment, CM mice clearly showed hemorrhage areas, which could be observed macro (Figure 1D,E) and microscopically (Figure 2A–C), as well as the presence of vascular obstruction areas (Figure 2C,D). On the other hand, mice protected against CM did not present cerebral damage, neither macro (Figure 1F,G) nor microscopically (Figure 2E,F).

Considering that curative treatment with alkaloids **4** and **10** was able to protect against CM, but did not prevent parasitic growth, in order to assess whether the treatment method could influence the effectiveness of drugs, the suppressive tests were performed. For this, four hours after infection, the animals were orally treated with the compounds for five days at the same doses as the curative treatment regimen (10 mg/kg). From the 4th dpi (last day of treatment), blood parasitemia, clinical score, and survival of the animals were followed.

As seen in Table 5, the parasitemia of mice infected and treated in a suppressive regimen with compound **10** was statistically lower than the untreated group on the 5th dpi (Tukey test, *p* < 0.05). However, the IMP potential was not sustained in subsequent days and on the 7th dpi, the group treated with the alkaloid **10** showed a reduction of almost 50% in IMP potential compared to 5th dpi. Even so, this compound can still be considered active at this point in the study. Compound **4**, despite having exhibited an IMP potential of 40% at 5th dpi, showed no statistically significant difference compared to the untreated but infected group. Compound **3** was not effective in inhibiting the parasite growth on a suppressive treatment regimen. This result suggests the total ineffectiveness of the compounds **3** and **4** in reducing parasitemia when administered in a suppressive regimen. 

The suppressive treatments with the compounds **3** and **10** were also able to inhibit the development of CM more efficiently than curative assay (Figure 3A–C). However, in the suppressive treatment regimen, compound **4** showed a reduction in the protection rate against CM compared to the curative experiment. Compound **4** continued to be significant, protecting 60% of the animals (Figure 3A). All animals belonging to the untreated control group (PbA) developed CM between the 6th and 11th dpi, evidenced by mean parasitemia of less than 10% (Figure 3B), mean clinical score ≤ 8 (Figure 3C), and presence of cerebral hemorrhage macroscopically observed (Figure 3D,E). 

Among the various classes of alkaloids, indole alkaloids stand out as the largest class found in nature. They are derived from the amino acid tryptophan, which, through decarboxylation, forms tryptamine and gives rise to six subclasses of indole alkaloids through varied biosynthetic routes, including β-carbolines and quinolines [66,67]. The importance of alkaloids in the treatment of malaria goes back a long way, since the discovery of the quinine, the first antimalarial drug [68]. Originating from the *Cinchona calisaya* species, quinine was isolated and characterized as an active substance, whose chemical structure is mainly composed of an alkaloid, which is responsible for its antimalarial activity [69,70]. Despite the known antimalarial activity of quinoline alkaloids, another subclass of indole alkaloids, the β-carbolines, has been highlighted since some studies have demonstrated its efficacy in vitro and in vivo against *Plasmodium* spp. [16,55,56,60,61,71,72,73,74]. 

Some β-carboline derivatives were in vitro active against both sensitive or multiresistant *P. falciparum* strains, showing low IC_50_ and important SI values [71,74]. In the in vivo trials, these studies used non-severe malaria induced by the *P. berghei* NK65 model, and an antiplasmodial activity was observed. The suppression of parasitemia found by Gorki et al. (2020), at 5th dpi, ranged from 40.49% to 91.19% at 10 and 100 mg/kg, respectively [74]. In our study, we found in a CM model for the compound **10** at a dose of 10mg/kg, a percentage of reduction of 82.07% and 67.92% at 5th dpi, in the suppressive and curative assays, respectively. These data suggest a better antimalarial potential to this compound, comparing to the others β-carboline alkaloids previously tested [72,74]. It is interesting to note that a derivative from antimalarial MMV008138 was efficacious in vivo mouse model of malaria, although this is the first report to describe the protection of β-carbolines against CM [56].

Compound **10** stood out against the other compounds being able to protect, at least 60%, the animals against the development of CM in both curative and suppressive regimens. Compared to the other tested compounds, the compound 10 presents significant structural changes, such as: (i) lacks of methyl ester substituent, present in compounds **3**–**6**, (ii) lacks of stereocenters, and (iii) presences of an aromatic ring; being these last two structural features absent in all other compounds of the series.

#### 3.3.4. Evaluation of NO Production

Considering that CM is related to an exacerbated inflammatory process [75,76], we sought to investigate whether compound **10** has anti-inflammatory activity by reduction of nitric oxide (NO). For this purpose, the in vitro NO production by murine peritoneal macrophages treated with compound **10** at different concentrations was investigated (Figure 4). However, to ensure that the NO production was the result of an inhibition of this pathway by the compound, and not by a reduction in the macrophage number due to a possible toxic effect of the compound against these cells, firstly, the MTT assay on peritoneal macrophages of BALB/c mice was performed. 

The results demonstrated that alkaloid **10** has no toxicity for the cells tested, since no statistical difference was observed in all tested concentrations (Figure 4A). After investigating the cell viability, the anti-inflammatory activity of **10** was evaluated through the dosage of NO (Nitrite equivalent). The results showed that compound **10** was able to reduce NO levels at all concentrations tested (*p* < 0.05) (Figure 4B). Furthermore, the concentrations of 150 µM and 74 µM were considered statistically similar to the basal group (*p* > 0.05) (Figure 4B).

Nitric oxide (NO) is a signaling molecule produced by a family of nitric oxide synthase (NOS) enzymes which acts in multiple functions in the body [77]. Physiologically, NO is produced at low concentrations (iNOS-2 to 200 nM), and under pathological conditions can reach micromolar concentrations [78]. In this context, its role is widely discussed and seems to be controversial, which may provide benefits or deleterious effects, depending on the exposure time, amount of production, and the biological site where it is released [79].

There are three isoforms of the NOS enzymes, but the inducible form (iNOS) is observed in the cell’s membrane with immune function, such as macrophages and microglia, whose expression is activated in inflammatory processes [77]. In parasitic diseases, such as malaria, NO has been described as an important molecule, being produced in large amounts by iNOS in response to the stimulus of the pathogen and pro-inflammatory cytokines [80,81].

Reactive oxygen and nitrogen species (ROS/RNS) are produced during CM, and the increase of these substances is associated with the development of CM in humans. Some authors suggest that among RNS, especially NO has a cytotoxic effect, being able to cross the blood–brain barrier and interferes with neurotransmission, releasing large amounts of glutamate during the excitotoxic process [7,82,83]. The result of this process is the releasing of ROS and NO, promoting neuronal death and thus contributing to cases of loss of consciousness or reversible coma [82,83]. Through the results found, it was possible to observe that compound **10** significantly reduced NO levels in stimulated macrophages, showing that it is a good candidate drug for inflammation control.

In order to understand how the compound **10** can lead to NO reduction, molecular docking on iNOS was carried out. Firstly, the redocking for the crystallographic ligand present in the enzyme (RMSD = 0.972) was performed (Appendix A), and this compound showed high affinity for the catalytic site (−9.9 kcal/mol), interacting with the Tyr 367 residue by hydrogen bonding (Appendix A). After this step, we performed in silico analysis for compound **10**, and the results showed that the compound **10** was able to interact with the Glu 371 and Arg 260 residues through hydrogen bonding, presenting a promising inhibition energy value (−9.6 kcal/mol) (Appendix A).

Inducible nitric-oxide synthase (iNOS) is a heme protein that requires tetrahydrobiopterin (H4B) for activity [84]. Among the active site residues, the Glu371 residue plays one of the most important roles, being critical for substrate binding in this enzyme [85]. Thus, the interaction of compound **10** with a residue from the catalytic site of iNOS may be the reason for its inhibition and also for the results found by the dosage of NO.

## 4. Conclusions

In summary, among all the tested compounds, we highlight compound **10** since: 1—in silico predictions suggest that this compound can interact with different chemotherapeutic targets of *P. falciparum*, such as PfFNR, PfPK7, PfGrx1, and PfATP6; 2—is able to inhibit both in vitro and in vivo parasite growth; 3—protected mice against CM development; and 4—showed a good potential for interaction with iNOS, leading to reduction of NO synthesis, which may be one of the pathways of protection against ECM. These results pointed the β-carboline derivatives as potential candidates to fight CM.

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
