# Peer review of "Rational-Based Discovery of Novel β-Carboline Derivatives as Potential Antimalarials: From In Silico Identification of Novel Targets to Inhibition of Experimental Cerebral Malaria"

_pathogens, 2022, doi:10.3390/pathogens11121529_

Round 1
Reviewer 1 Report
With respect to the manuscript titled: ’’Rational-based discovery of novel β-carbolines derivatives as potentials antimalarials: from in silico identification of novel targets to inhibition of experimental cerebral malaria’’.
The manuscript describes design, synthesis and biological evaluation of seven synthesized β-carbolines for antimalarial activity. The structures were selected as a result of in silico screening of 20 designed β-carbolines against a pool of Plasmodium falciparum molecular targets, the Brazilian Malaria Molecular Targets (BRAMMT). Synthesized derivatives showed modest activity against cloroquine-resistant P. falciparum strain W2 and good selectivity for the parasite in comparison to the human pulmonary fibroblast cells. After predicting favorable physicochemical and pharmacokinetic properties of selected derivatives, the compounds were subjected to in vivo evaluation. One compound showed parasite growth inhibition on the 5th day post-infection for the both curative (67.9 %) and suppressive tests (82%). The most important, this compound protected the mice (>65%) against the development of the experimental model of cerebral malaria, probably by reducing NO synthesis.
In my opinion, the manuscript should be published after some revisions provided below.
1. I think that the research is well designed, systematically carried out, and the conclusions are supported by the obtained results. In my opinion, the weakness of the manuscript is insufficient emphasis of the most important results - inhibition cerebral malaria development in mice for the first time with β-carbolines. You should point that out in Highlights and Conclusion, too.
2. Highlights
- Possible interactions with different enzymes are only established in silico. So, I think that you should not claim that those compounds may cause the parasite death by inhibiting different enzymes
- The tested compounds showed the same or even less activity against P. falciparum CQ-resistant strain W2 in comparison to CQ, I recommend changing „high efficacy“ into „eficacy“ or „modest activity“.
3. Graphical Abstract
- Please provide higher quality image for graphical abstract, enhance the resolution
4. Experimental design
- I recommend putting this section after Results and Discussion, because I think that the structures of compounds should be introduced earlier in the text (Scheme 1)
- Line 112: Change “was added dropwise concentrated sulfuric acid (2,81 mmol)” into “concentrated sulfuric acid (2.81 mmol) was added dropwise”.
- Line 117: change “compound 2” into “compound 2”. Use bold numbers for synthesized compounds in the text.
- Line 123: The missing words in the sentence “To a mixture of L-tryptophan methyl ester or tryptamine (0.30 mmol), aromatic benzaldehyde derivatives (0.90 mmol) in methylene chloride (35 mL) TFA (0.30 mmol) was then dropwise” should be added.
- Line 131 and all experimental procedures: use “anhydrous Na2SO4” instead of “Na2SO4”
- Line 184: change “was added 0.80 mmol potassium permanganate” into “potassium permanganate (0.80 mmol) was added”.
- Line 197: change “07 compounds” into “seven compounds”
- Line 204 and elsewhere: There should be consistency with the use of italic style for writing parasite strain names and “in vitro“, “ in vivo” throughout the text.
- Line 209: change “at 37
º C” into “at 37 °C”
- Line 221: change “number of hydrogen bond acceptors (HBA) and were predicted” into “and number of hydrogen bond acceptors (HBA) were predicted”
- Line 226: change “Evaluation in vivo antimalarial activity” into “Evaluation of in vivo antimalarial activity”
- Line 234: ad libitum should be italic
5. Results and Discussion
- Table 1: The models used in experiment are described in the text, but I recommend short description right below the table 1
- Line 369: When mentioning compounds, it is necessary to state that they are on the Scheme in the text, not only to refer to Supplementary file, for example : compound 5 (Figure S2D)
- Line 380: Reformulate “inhibition of the Asp301 residue”
- Table 2: Remove ±SD from WI-26-VA4 IC50 ± SD (μM)
- Line 409: delete “for many researchers”
- Line 446: The abbreviation CQ for chloroquine should be introduced when chloroquine is first mentioned and then used further through the text
- Line 462: The statement: “Likewise, no difference was observed in the proportion of parasites circulating in the groups treated with the alkaloids 4, 6 and 10 in relation to the standard antimalarial at 5th dpi (Tukey Test, p > 0.05)” should be checked and corrected.
- Line 478: change “MC”into “CM”
- Line 547: change “derivate” into “derived”
- Line 558: change “IS” into “SI”
- Line 569: I think that the structure difference between compound 10 and other tested compounds is significant and that should be also noted
- Line 571: the statement “CM is related to an exacerbated inflammatory process” needs reference
- Line 593: change “150” into “150 µM”
Reviewer 2 Report
Dear authors,
The manuscript titled "Rational-based discovery of novel beta-carbolines derivatives as potentials antimalarials..." was an interesting work on the screening of drugs for treatment of cerebral malaria, but I have some concerns as follows:
1. the sources of parasites of Pf and Pb?
2. the Pf W2 strain was the chloroquine-resistant strain, what about the drug sensitivity of Pb? and did the strain with some sensitivity or resistance affect the evaluation of the drug screening?
3. pls clarify the full name of ECM, experimental cerebral malaria?
4. pls add more contents to provide the reason to focus on the study of macrophage and NO. Otherwise, the overall structure of the article appears more abrupt and lacks continuity and integrity.
Round 2
Reviewer 2 Report
Dear authors,
Thank you for your revision.
Best,